# Life Prediction of Rolling Bearing Based on Optimal Time–Frequency Spectrum and DenseNet-ALSTM

**DOI:** 10.3390/s24051497

**Published:** 2024-02-26

**Authors:** Jintao Chen, Baokang Yan, Mengya Dong, Bowen Ning

**Affiliations:** School of Information Science and Engineering, Wuhan University of Science and Technology, Wuhan 430081, China; chenjintao@wust.edu.cn (J.C.); ybk870610@126.com (B.Y.); dongmengya@wust.edu.cn (M.D.)

**Keywords:** signal reconstruction, generalized S-transform, rolling bearing, life prediction

## Abstract

To address the challenges faced in the prediction of rolling bearing life, where temporal signals are affected by noise, making fault feature extraction difficult and resulting in low prediction accuracy, a method based on optimal time–frequency spectra and the DenseNet-ALSTM network is proposed. Firstly, a signal reconstruction method is introduced to enhance vibration signals. This involves using the CEEMDAN deconvolution method combined with the Teager energy operator for signal reconstruction, aiming to denoise the signals and highlight fault impacts. Subsequently, a method based on the snake optimizer (SO) is proposed to optimize the generalized S-transform (GST) time–frequency spectra of the enhanced signals, obtaining the optimal time–frequency spectra. Finally, all sample data are transformed into the optimal time–frequency spectrum set and input into the DenseNet-ALSTM network for life prediction. The comparison experiment and ablation experiment show that the proposed method has high prediction accuracy and ideal prediction performance. The optimization terms used in different contexts in this paper are due to different optimization methods, specifically the CEEMDAN method.

## 1. Introduction

In today’s manufacturing industry, rolling bearings play a crucial role as integral components of various mechanical devices. Incidents involving component failures in these bearings can lead to production accidents, and in more severe cases, even result in the loss of lives. Therefore, timely monitoring of the remaining service life of bearings is essential and holds significant importance in the production processes of enterprises. However, due to the operating conditions of rolling bearings, they often generate considerable noise, and fault signals may not be readily apparent during their operation. This inherent challenge has resulted in inaccurate predictions of bearing life. Consequently, improving the accuracy of predicting the remaining service life of bearings, particularly in high-noise environments, is crucial. This improvement enables the precise monitoring of the operational status of bearings, making it of paramount significance [1].

The current methods for predicting bearing life can be broadly categorized into three types. (1) There are prediction methods based on physical models. For instance, Lu et al. proposed an intelligent prediction approach for rolling bearing fault feature extraction based on physics. This method utilizes Realized Volatility (RV) and Wavelet Neural Network (WNN) to predict the remaining life of bearings. It overcomes the challenges of predicting bearing faults under various degradation modes [2]. (2) Prediction methods based on probability statistics constitute the second category. For example, Kundu et al. introduced a Weibull accelerated failure time regression model for predicting the remaining service life of bearings under multiple operating conditions. This model enables the prediction of bearing life under various operating conditions [3]. (3) The third category involves prediction methods based on data-driven approaches. For instance, Guo et al. proposed a bearing life prediction method based on a hybrid data-driven approach. This method combines grey models, complete ensemble empirical mode decomposition, and relevance vector machines, achieving a relatively ideal level of prediction accuracy [4].

Due to the increasing complexity of mechanical equipment structures, it is challenging to establish accurate physical models for failure mechanisms. Consequently, the obtained solutions are only approximations, and the modeling process cannot account for various influencing factors. Obtaining physical models for the working processes of many pieces of equipment is often uneconomical or even impossible. While statistical models for remaining life prediction can effectively forecast the remaining life of mechanical equipment, most methods assume that the degradation model is known. They then estimate the model parameters using state monitoring data or environmental data and, finally, use the model to predict remaining life. However, in practical engineering applications, the degradation model for the performance of mechanical equipment is often unknown, and different types of equipment may have different degradation models. If the degradation model is chosen improperly, it can significantly impact the accuracy of the remaining life predictions. On the other hand, data-driven prediction methods can better collect and process data from rolling bearings during operation. By constructing deep learning networks tailored to different types of data, these methods can uncover performance degradation information embedded in various types of data, representing a trend in the development of bearing life prediction [5].

Shi proposed a bearing life prediction method based on a multilayer graph-embedded Extreme Learning Machine (ELM). By integrating a graph embedding framework, they constructed a novel embedded graph ELM autoencoder, significantly reducing training time and improving prediction efficiency [6]. Guo introduced a method for predicting the remaining useful life of rolling bearings based on EMD-RISI-LSTM. This approach utilizes the RISI method to select Intrinsic Mode Function (IMF) components with more degradation characteristics to train the Long Short-Term Memory (LSTM) model, thereby enhancing accuracy and robustness under different operating conditions [7]. Liu proposed an improved method for bearing life prediction using a modified Support Vector Regression (SVR) model. They introduced a method based on a Hybrid Kernel Function–Support Vector Regression (HKF–SVR) to make predictions. This approach allows for the extraction of potential source features, accurately reflecting performance degradation from mixed vibration signals [8]. Wang proposed a rolling bearing life prediction model that combines Convolutional Neural Network (CNN) and Long Short-Term Memory (LSTM). This model can automatically extract fault features from input time domain signals, addressing issues such as inaccurate fault feature extraction [9]. The abovementioned networks are all based on time domain signal prediction. Due to the nonstationary nature of bearing fault signals, relying solely on time or frequency domain perspectives can lead to insufficient accuracy in automatic feature extraction by the prediction network. This limitation makes it challenging to effectively eliminate irrelevant features from the signal.

Under the influence of strong background noise, the fault signals of rolling bearings are relatively weak compared to background noise. This makes the extraction of fault features from the signal challenging. Therefore, the key to extracting signal features lies in denoising and enhancement, especially in the presence of strong background noise. Lei proposed a fault diagnosis method for rolling bearings using SSA-IWT-EMD [10]. Chen et al. developed a fault diagnosis method for subway train rolling bearings based on SSA-VMD [11]. This method requires optimizing the number of modes in the VMD decomposition, and different functions are often used for optimization, yielding varying results. Therefore, selecting a cost function is also a crucial consideration in this process.

Compared to time domain and frequency domain signals, the time–frequency spectrum can more intuitively present the characteristics of bearing vibration signals. Yang proposed a method for identifying faults in rolling bearings using the Hilbert time–frequency spectrum. The collected raw time domain signals are transformed into two-dimensional time–frequency images through Hilbert transformation, providing a better representation of local time–frequency features in the signal [12]. However, this method has limitations in the process of predicting the lifespan, where a fixed window width in the window function may lead to the loss of some high-frequency information, making it less effective in accurately characterizing the vibration features of bearings.

Therefore, addressing the aforementioned issues, a rolling bearing life prediction method is proposed, which combines the optimal time–frequency spectrum with a DenseNet-ALSTM network. The optimal generalized S-transform time–frequency spectrum, obtained through signal reconstruction enhancement and optimization using the particle swarm optimization algorithm, significantly highlights the time–frequency characteristics of fault signals in the time–frequency domain. Its anti-interference capability is notably superior to that of single time domain or frequency domain signals. This method overcomes the limitations of the constant time–frequency resolution in the STFT time–frequency spectrum due to an unchanging window width and the weakness of wavelet transform in decomposing one-dimensional signals into the time-scale domain only. Additionally, the signal reconstruction method proposed in this study does not require parameter optimization and exhibits a noticeable reconstruction effect, and the DenseNet-ALSTM network used in the experiments effectively extracts fault features from the time–frequency spectrum through DenseNet and self-attention modules. This enables the network to eliminate redundant features in the input, avoiding issues such as inaccurate feature extraction caused by the prediction network. The bearing data used in this study were generated from bearing simulation test benches and do not take into account the influence of resonance phenomena produced at the mechanical level.

## 2. Theoretical Background

### 2.1. DenseNet-ALSTM Network

The network structure used in this paper consists of a DenseNet (Densely Connected Network), attention mechanism, and Long Short-Term Memory (LSTM) network. The DenseNet network is a novel Convolutional Neural Network (CNN) architecture proposed by Gao Huang et al. in 2017. Unlike the ResNet network, which combines features through summation across layers, DenseNet employs a connectivity pattern to merge features. This approach not only addresses issues like vanishing or exploding gradients but also, to some extent, reduces the number of network parameters. As a result, the network becomes more trainable. Additionally, DenseNet resolves overfitting problems that may arise when the training dataset is limited [13]. The specific network structure is as follows:Dense Block: The dense block is composed of multiple layers, and each layer is densely connected to achieve the combination of features.(1)Dense Connectivity: DenseNet introduces a novel connection approach where the output of each layer is densely connected to the outputs of all preceding layers along the channel dimension. This process aims to reuse features and pass them as inputs to subsequent layers, termed dense connectivity. This method significantly enhances the flow of information between layers, facilitating deep feature extraction.(2)Composite Function: The nonlinear transformation function *H_l_* is defined as a composite function consisting of three consecutive operations. This composite function comprises a batch normalization function (BN), an activation function (ReLU), and a 3 × 3 convolution function (Conv).Transition Layer: To achieve downsampling of the feature maps output by dense blocks, a network structure introduces a transition layer that connects two adjacent dense block units. The transition layer mainly consists of a batch normalization operation, a ReLU activation function, a 1 × 1 convolution layer, and a 2 × 2 average pooling layer.

Above all, the specific network structure of DenseNet is illustrated in Figure 1:

After extracting data features through the DenseNet network, it is necessary to employ Long Short-Term Memory (LSTM) networks to predict the remaining useful life of the bearings. The fundamental unit of an LSTM consists of a forget gate, an input gate, and an output gate. LSTM networks are an improvement over recurrent neural networks (RNNs), addressing the issue of vanishing gradients encountered in traditional RNNs during runtime. They are suitable for sequential data and exhibit a certain degree of memory effect.

Simultaneously, in this network architecture, to filter the features extracted by the DenseNet network and enhance the accuracy of the predictions, a self-attention module has been added on top of the DenseNet network. Its primary function is to improve the accuracy of the predictions by utilizing attention mechanisms during the deep feature extraction process of DenseNet. The attention module can eliminate ineffective features and redundant information, thereby filtering out irrelevant data from the feature set. This, in turn, contributes to an enhancement of the prediction accuracy.

### 2.2. Generalized S-Transform

When there is a bearing vibration signal *x*(*t*), the expression for its generalized S-transform is given by
(1)$$S\left(\tau ,ƒ\right)=\int_{-\infty }^{+\infty }x(t)\frac{|ƒ{|}^{p}}{\sqrt{2\pi }}e^{\frac{-{\left(\tau -t\right)}^{2}{ƒ}^{2}}{2}}e^{-j2\pi ft}dt$$
where *τ* is the time factor, *f* is the frequency factor, and *p* is the tuning factor. Parameter *p* adjusts the width of the Gaussian window, thereby enhancing the time–frequency concentration of the S-transform. When *p* = 1, the generalized S-transform reduces to the standard S-transform [14].

### 2.3. Snake Optimization (SO) Algorithm

The SO algorithm is a novel optimization algorithm that mimics the unique mating behavior of snakes. The algorithm consists of two phases: global exploration and local exploitation. It leverages the characteristics of snake behavior, which are influenced by external temperature and the variety of available food, to rapidly explore the search space, offering high efficiency and strong accuracy [15]. The algorithmic flowchart is depicted in Figure 2.

## 3. Life Prediction of Rolling Bearing Based on Optimal Time–Frequency Spectrum and DenseNet-ALSTM

During the process of predicting the life of rolling bearings, the collected vibration signals are often contaminated with significant background noise. This leads to the submersion of the true features of bearing vibration signals, making it challenging to accurately extract these features. Consequently, establishing specific models for the degradation state of bearings becomes difficult. Additionally, traditional data-driven life prediction methods are limited by the nonstationary nature of time domain signals. Moreover, conventional Recurrent Neural Network (RNN) models are prone to the exploding gradient problem when dealing with long-term predictions. To address these challenges in bearing life prediction, many experts have proposed various solutions.

To address the issue of excessive background noise, Zhao et al. proposed a moving average coarse-graining method for denoising bearing vibration signals. This approach significantly enhances the interference resistance of degradation indicators, improving stability and prediction accuracy [16]. Zhuang et al. introduced a denoising method for bearing signals based on VMD sample entropy [17]. Meng et al. presented a denoising method for bearing signals based on the optimal MOMEDA method [18]. All of these methods can denoise the original bearing signals, thereby improving prediction accuracy. The Mean Squared Error (MSE) indicators for these methods are presented in Table 1.

Table 1 shows that denoising the original signals significantly improves the accuracy of the predictions, thereby addressing the challenge of extracting the degradation state model of the bearings.

Furthermore, due to the advantages of time–frequency spectral analysis, such as being more comprehensive, finer, having better visualization, and improved fault diagnostic capabilities compared to time domain or frequency domain signal analysis, Ding et al. proposed a method using continuous wavelet transform to transform the collected raw time domain signals into two-dimensional time–frequency images. This approach aims to better capture the local time–frequency features in the signal [19]. Liu et al. introduced a bearing remaining life prediction model based on STFT-CNN. Prior to inputting the signal into the CNN model, the original signal undergoes STFT transformation, translating the one-dimensional time series signal into the time–frequency domain, resulting in improved prediction performance [20]. This allows for a more accurate construction of the bearing degradation model, thereby enhancing the accuracy of the bearing life predictions.

Finally, to address the challenges of gradient explosion and the resulting lower prediction accuracy in traditional data-driven methods, many experts have proposed effective solutions by introducing novel network architectures. For example, Xiao et al. introduced a bearing remaining useful life prediction method based on a two-dimensional attention residual network. This method involves constructing an attention residual network with triple features as the input for predicting the remaining useful life of the bearings. Channel attention and spatial attention are concatenated into the residual connections of the residual neural network, creating a new attention residual module. This allows for the newly constructed deep learning network to better focus on subtle changes in the orientation state, leading to a significant improvement in prediction accuracy [21]. Wang et al. proposed a Bayesian Large Kernel Attention Network for bearing remaining useful life (RUL) prediction and uncertainty quantification. This network incorporates uncertainty quantification, long-range correlations, and channel adaptability into the attention mechanism, effectively extracting degradation features and enhancing the accuracy of RUL predictions. Notably, it achieves uncertainty quantification not only in RUL prediction but also consistently outperforms traditional prediction networks in terms of performance [22].

Based on the theoretical approaches mentioned above, this study employs the Teager energy operator, a superior denoising method compared to conventional techniques, to extract the impact envelope of time domain signals. This method better reflects the time–frequency spectrum, specifically the optimal generalized S-transform, which serves as the input for the model. Consequently, a life prediction method based on the optimal time–frequency spectrum and DenseNet-ALSTM is proposed. The approach consists of three main steps: (1) Enhancing impacts using a signal reconstruction method to obtain the strengthened signal. (2) Optimizing the time–frequency spectrum of the enhanced signal using the SO algorithm to acquire the optimal time–frequency spectrum. (3) Utilizing the optimal time–frequency spectra of all samples as an input for life prediction through the DenseNet-ALSTM network. This approach addresses issues such as the submergence of fault features in the original signal, inaccurate feature extraction, low prediction accuracy, and the occurrence of gradient explosions in the network. The flowchart of the life prediction method in this study is illustrated in Figure 3.

### 3.1. Impact Enhancement

Rolling bearings operate under complex conditions, and vibration signals are often contaminated with strong background noise, thereby masking the fault impact features within the vibration signal. The signal reconstruction method proposed in this paper, due to its proportionality to the square of amplitude and frequency, effectively highlights the impact components. Traditional signal reconstruction methods are only suitable for demodulating impact signals and may not perform well under strong noise interference. In this study, the signal reconstruction method is employed for signal preprocessing. While enhancing the impact, it preserves the time–frequency characteristics of the signal.

#### 3.1.1. Extract the Signal Envelope

This paper takes advantage of the Teager energy operator’s ability to highlight the impact features of time domain signals and extract the envelope of the signal. When there is a continuous time signal *x*(*t*), the envelope of the signal is given by [16]
(2)$$\Psi \left[x(t)\right]={x^{\prime }}^{2}(t)-x(t)x^{\prime \prime }(t)$$
where *x*′(*t*) is the first derivative of the signal *x*(*t*) concerning time *t*, and *x*′′(*t*) is the second derivative of the signal *x*(*t*) to time *t*.

If there is a simulated time domain signal, the equation appears as follows:(3)$$x(t)=e^{-\frac{\lambda^{2}{\left(t-u\right)}^{2}}{2}}\cos(2\pi ft)$$
where the scaling factor *λ* is 64, the sampling frequency is 1200 Hz, the number of sampled points is 2400, and the signal frequency *f* is 70 Hz. The time domain plot is illustrated in Figure 4.

Therefore, according to Equation (2), the Teager energy operator envelope of the simulated time domain signal is
(4)$$\Psi \left[x(t)\right]=4\pi^{2}f^{2}e^{-\lambda^{2}{\left(t-u\right)}^{2}}$$

Moreover, the Teager energy operator envelope plot of the signal *x*(*t*) is depicted in Figure 5.

From the graph, it can be observed that, after the calculation with the Teager energy operator, the impulses in the time domain signal were extracted. Furthermore, the amplitude of the impulses was significantly enhanced, serving to highlight the fault characteristics of the time domain signal.

#### 3.1.2. Envelope Denoising

When the envelope signal is heavily disturbed by noise, it becomes challenging to separate the noise and impulse components. Therefore, it is necessary to perform high-frequency denoising on the signal envelope while preserving the low-frequency envelope components. The specific process of the envelope denoising method in this paper is as follows:(1)Signal decomposition. Let the simulated signal with noise be *X*(*t*). First, use CEEMDAN to decompose the vibration signal *X*(*t*).

To address issues like mode mixing in the EMD algorithm, Torres et al. proposed an improved algorithm based on EMD called the Complete Ensemble Empirical Mode Decomposition with Adaptive Noise (CEEMDAN). This method incorporates auxiliary noise-containing Intrinsic Mode Function (IMF) components obtained after EMD decomposition. It conducts an overall averaging calculation right after obtaining the first-order IMF component, deriving the final first-order IMF component. Subsequently, it repeats the same procedure for the residual part, effectively resolving the transfer of white noise from high to low frequencies. This approach mitigates mode mixing phenomena [23]. Its primary principles are as follows:

Step 1: Consider a time series signal with white noise denoted as *X*(*t*), where *θ^i^*(*t*) represents the white noise sequence, *ε*_0_ is the standard deviation of the white noise sequence, and *i* = 1, 2, 3, …; N corresponds to the number of experimental trials. Therefore, the signal constructed in this study is as follows:(5)$$X(t)=x(t)+\varepsilon_{0}\theta^{i}(t)$$

Step 2: Perform EMD decomposition on the constructed signal *X*(*t*) and obtain the *IMF* components from the decomposition. Calculate the average of these decomposed *IMF* components to derive *IMF*_1_:(6)$$IMF_{1}(t)=\frac{1}{N}\sum_{i=1}^{N}IMF_{1}^{i}(t)$$

Step 3: Compute the first residual component:(7)$$r_{1}(t)=x(t)-IMF_{1}(t)$$

Step 4: By introducing white noise *θ^i^*(*t*) to the residual component *r*_1_(*t*) and performing another round of decomposition, a set of *IMF* components is obtained. Calculate the average to derive *IMF*_2_:(8)$$IMF_{2}(t)=\frac{1}{N}\sum_{i=1}^{N}E_{1}\left\{r_{1}(t)+\varepsilon_{1}E_{1}\left[\theta^{i}(t)\right]\right\}$$
where *E_j_*() represents the *j*-th order component.

Step 5: Compute the second residual component:(9)$$r_{2}(t)=x(t)-IMF_{2}(t)$$

Step 6: Repeat the above steps until the conditions for EMD are no longer satisfied and terminate the decomposition. At this point, you have obtained *k* IMF components.

The time domain plot of signal *X*(*t*) is shown in Figure 6. It can be observed that, due to the overlay of white noise, the vibrational impacts of the signal *x*(*t*) are completely imperceptible. Additionally, the fault characteristics in the signal are also masked by Gaussian white noise.

Figure 7 shows the envelope of the energy operator extracted using the Teager energy operator. It can be observed that the impacts in the envelope signal are also covered by noise, and the extracted impact features are not distinctly apparent.

Figure 8 displays the nine IMF components obtained through CEEMDAN decomposition. From the images, it can be inferred that the envelope spectrum impacts are more pronounced after the decomposition and denoising process. The signal is less affected by noise, making it relatively easy to extract the envelope impacts of the signal. This lays a solid foundation for subsequent signal reconstruction.

(2)By calculating the correlation between each IMF component and the original signal, the IMF component most suitable for Teager energy operator signal reconstruction is selected. The implementation steps are as follows:

Step 1: Calculate the kurtosis values for each IMF component obtained through decomposition and normalize the values to obtain *K_i_*. *K_i_* represents the kurtosis value of the *i*th IMF component.

Step 2: Calculate the correlation between each IMF component obtained through decomposition and the original signal and normalize the correlation values to obtain *γ_i_*. The calculation formula is as follows:(10)$$\gamma_{i}=(x(t)-h_{i}(t))*A_{i}$$
where *x*(*t*) represents the signal before decomposition, *h*_*i*_(*t*) stands for the power signal of the *i*th IMF component obtained through Hilbert transform, *Aᵢ* denotes the amplitude of the *i*th IMF component, and (∗) denotes the convolution operation.

Step 3: Calculate the IMF selection index *βᵢ*, defined by the following formula:(11)$$\beta_{i}=\frac{1}{2}*K_{i}+(1-\lambda )\gamma_{i}$$
where *λ* is a constant, and in this context, it is chosen to be 0.5. *K_i_* represents the kurtosis value of the *i*th IMF component, and *γ_i_* denotes the correlation between the *i*th IMF component and the original signal. A higher value of the selection index *βᵢ* indicates a higher correlation of the IMF component with the original signal.

The IMF component with the highest correlation after the CEEMDAN decomposition of signal *X*(*t*) is depicted in Figure 9.

(3)Signal reconstruction. The high-frequency noise in the optimized envelope signal is suppressed. Multiplying it by the original signal yields the reconstructed fault signal. This reconstructed signal not only amplifies the impulsive components but also captures the time–frequency characteristics of the enveloped impulsive components. The computational formula is as follows:

(12)$$x_{a}(t)=\Psi_{a}\left[X(t)\right]*X(t)$$
where Ψ*_a_* represents the optimized envelope signal, and *X*(*t*) is the original signal. As shown in Figure 10, the simulated signal after reconstruction using the Teager energy operator exhibits that the reconstructed signal has faithfully reproduced the impulsive characteristics of the original noise-free signal. Furthermore, the amplitude of the impulsive components has doubled, eliminating the influence of Gaussian white noise.

### 3.2. Time–Frequency Optimization

The generalized S-transform time–frequency spectrum allows for the analysis of signals in both the time and frequency domains. It can comprehensively extract the features of the input signal into the time–frequency spectrum, exhibiting significantly better anti-interference capabilities compared to singular time or frequency domain signals. Within the generalized S-transform, there exists an exponent parameter, denoted as *p*, which adjusts the width of the window function. This parameter controls the time–frequency concentration of the time–frequency spectrum, influencing its display characteristics. To achieve optimal prediction results, it is necessary to extract the best time–frequency spectrum. Therefore, the SO algorithm is introduced to optimize the value of *p*. The specific steps are as follows:(1)Initialize the parameters of the SO algorithm, such as population size, number of iterations, etc. Additionally, constrain the optimization range of parameter p to be within the range of (0, 1) for seeking optimization.(2)Apply the generalized S-transform to the preprocessed bearing signal to obtain its time–frequency spectrogram after the generalized S-transform.(3)Use the time–frequency concentration function *M*(*p*) as the fitness function for the optimization of the SO algorithm. A smaller value of this function indicates a better time–frequency concentration. By computation, the fitness function’s value can be obtained. The *p*-value corresponding to the minimum value of the function represents the optimal *p*-value for the current iteration.

The employed time–frequency concentration function is
(13)$$M(p)=(\sum_{n=1}^{N}\sum_{k=1}^{N}|S_{x}^{p}(n,k){|}^{\frac{1}{q}}{)}^{q}$$

(4)Iterate and continuously optimize the positions of the population to obtain a new optimal solution.(5)When the optimization results reach the optimal solution, the optimization process is concluded. Finally, output the optimal parameter *p*-value for the generalized S-transform. Obtain the optimal time–frequency spectrogram using the generalized S-transform.

As shown in Figure 11, the specific flowchart for optimizing the time–frequency spectrum of the signal is illustrated.

As shown in Figure 12, the optimal time–frequency spectrum of the noise-free simulated signal obtained through time–frequency spectrum optimization is presented. It can be observed that the impulsive characteristics of the signal are fully displayed in the time–frequency domain.

As shown in Figure 13, the optimal time–frequency spectrum of the noise-added simulated signal *X*(*t*) is presented. It can be observed that compared to Figure 12, the fault characteristics of the simulated signal after adding noise have been completely covered by noise, and the features are no longer prominent.

As shown in Figure 14, the optimal time–frequency spectrum image after reconstructing the signal using the optimal IMF components of the signal *X*(*t*) is presented. Compared to the optimal time–frequency spectra of the noise-added simulated signal and the noise-free simulated signal, it can be observed that the reconstructed signal’s time–frequency spectrum has removed the influence of noise. The fault characteristics of the signal are clearly manifested in the time–frequency spectrum image.

### 3.3. Bearing Life Prediction

In this paper, the DenseNet-ALSTM network is employed for life prediction. This network utilizes DenseNet architecture and self-attention modules to extract fault features from the time–frequency spectrum comprehensively. It effectively eliminates redundant features from the input, thereby avoiding issues such as inaccurate feature extraction caused by the prediction network. The prediction steps are as follows:(1)The vibration signals of the rolling bearings throughout their entire life cycle were enhanced for feature enrichment, reconstructed, and then optimized for each sample using the SO algorithm to obtain the optimal time–frequency spectra. This process allowed for the construction of an optimal time–frequency spectral dataset for the entire life cycle of the bearings.(2)Lifespan labels were established, with the time from the start of the bearing operation to complete degradation considered as the bearing’s entire life cycle. The sampling points at the initial run were labeled as 1, and those at complete degradation were labeled as 0. In this way, the entire life cycle trend of the bearing was defined within the range of (0, 1), serving as the labels for training the dataset.(3)The dataset was randomly divided into a training set and a test set. The training set was fed into the DenseNet network for adaptive feature extraction, enabling deep exploration of the time–frequency spectrum image features. A health indicator model was constructed using the extracted indicators, and these indicators were input into the ALSTM network model for training. Subsequently, the remaining lifespan of the bearings in the test set was predicted.

## 4. Bearing Signal Simulation Experiment Verification

The vibration signals which occurred when rolling bearings experienced faults were simulated through an experimental simulation. The assumed fault signal is
(14)$$h(t)=e^{\left(-c*t_{0}\right)}*\cos(2\pi f_{n}t_{0})$$
where the resonance frequency (*f_n_*) is 2500 Hz, the damping coefficient (*C*) is 500, the number of sampling points is 2048, the fault frequency is 35 Hz, and the sampling frequency is 12,000 Hz. The corresponding time domain graph is shown in Figure 15.

### 4.1. Signal Reconstruction

Due to the fact that in practical applications, the impact signals of faulty bearings are often submerged in strong noise, we simulated bearing vibration signals in this scenario. Gaussian white noise with a signal-to-noise ratio (SNR) of −5 dB to −10 dB was, respectively, added to the signals to generate two sets of noisy simulation signals, *S*_1_ and *S*_2_. Figure 16 illustrates the simulated signals with noise.

It can clearly be observed that as the SNR decreases, the impact components of the fault signal are significantly masked by noise, and the impact is essentially invisible.

Signal reconstruction can extract the bearing fault impact masked by noise. Due to the higher noise level in low-SNR conditions, it is necessary to denoise the signal envelope, retaining only the extracted impact components. This ensures a more accurate preservation of the impact in the original signal. Figure 17 shows the Teager Energy Operator envelopes of signals *S*_1_ and *S*_2_, decomposed through CEEMDAN, with the IMF component that has the highest correlation with the original signal.

It is clearly visible in the figure that after CEEMDAN decomposition, the six fault impacts in the signal envelope have been successfully extracted, and they are notably effective even in low-SNR conditions.

Utilizing the aforementioned signal envelopes and applying Equation (12) for signal reconstruction results in the generation of feature-enhanced signals [24]. Figure 18 depicts the enhanced signals after signal reconstruction.

It can be observed that after reconstruction, the fault impacts in the signal are more pronounced compared to before processing, making it more conducive for extracting the fault characteristics from the signal.

To present the effects of the proposed signal reconstruction method more intuitively, Table 2 shows the SNR of the signals before and after reconstruction. It can be observed that the SNR of the reconstructed signals is significantly improved.

### 4.2. Time–Frequency Optimization

The generalized S-transform can transform the reconstructed signal from the time domain to the time–frequency domain. As per Equation (1), the result of the generalized S-transform is the inner product of the fault signal *h*(*t*) and the Gaussian window function *w*(*t*, *f*) [24]:(15)S(t,f)=<h(t),w(t,f)>

Furthermore, by making a comparison with Equation (13), it can be deduced through calculations that the time–frequency concentration function *M*(*p*) reaches its minimum value only when the average energy *S* attains its maximum. At this point, the fault features displayed in the time–frequency spectrum are most pronounced.

To obtain the optimal time–frequency spectrum, this paper employs the SO algorithm to optimize parameter p. The time–frequency concentration is used as the fitness function, where a smaller time–frequency concentration value indicates a better concentration and higher energy in the time–frequency spectrum. Figure 19 shows the variation in the time–frequency concentration with parameter *p* for the noise-free simulated signal, and Figure 20 illustrates the variation in average energy in the time–frequency spectrum with parameter *p* for the noise-free simulated signal.

After optimization using the SO algorithm, the optimal value for *p* is found to be 0.8865, which aligns with the minimum value in Figure 19 and the maximum value in Figure 20.

Compared to traditional optimization algorithms, the SO algorithm has higher efficiency and can achieve optimal fitness values in fewer iterations. Figure 21 depicts the convergence curves of the Whale Optimization Algorithm (WOA) and particle swarm optimization (PSO) and SO algorithms in optimizing the time–frequency spectrum.

From Figure 21, it is evident that the SO algorithm requires fewer iterations and is less prone to being trapped in local optima compared to the other two algorithms. Therefore, it exhibits higher optimization efficiency. Moreover, Figure 22 illustrates the optimal time–frequency spectrum obtained through optimization by the WOA and PSO and SO algorithms.

Based on the above experimental results, it is evident that the SO algorithm achieves the best optimization for the time–frequency spectrum with higher efficiency. Consequently, after reconstructing signals with two different signal-to-noise ratios, the SO algorithm was applied for the optimization of the time–frequency spectrum. Figure 23 depicts the optimal time–frequency spectra of simulated signals with varying SNRs.

From Figure 23, it can be observed that under high background noise, the optimal time–frequency spectrum obtained through signal reconstruction and SO optimization clearly reveals fault features, no longer submerged in the noise. This is advantageous for extracting features that contribute to life prediction.

## 5. Actual Data Analysis

The actual dataset used in this experiment is the PHM-2012 bearing full-life dataset released by IEEE in 2012. This dataset consists of 17 sets of bearing data, covering the entire operational lifespan of the bearings from the beginning of operation to failure. The data are sampled at a frequency of 25.6 kHz, with a sampling time length of 0.1 s, and each recorded sample consists of 2560 data points. The dataset includes three different operating conditions: Condition 1 at 4000 N and 1800 rpm, Condition 2 at 4200 N and 1650 rpm, and Condition 3 at 5000 N and 1500 rpm [25]. Figure 24 shows the vibration signal over the entire lifespan of bearing 1-1.

From Figure 24, it can be observed that the amplitude of the vibration signal increases as the bearing operates. In the early stages, the signal indicates normal operational conditions. In the middle stage, the bearing begins to degrade, and the amplitude of the signal increases with operating time. In the later stages, there is a sudden doubling of the amplitude, indicating that the bearing has failed at this point [26,27].

### 5.1. Data Preprocessing

To enhance the characteristics of the bearing vibration signal and reduce noise, facilitating better feature extraction, a signal reconstruction method was applied to all the sampled data points from the 17 sets of bearing datasets for feature enhancement. To demonstrate the effectiveness of this method on real signals, Figure 25 illustrates the last set of signals from bearing 3-3 (the actual fault signal) selected from the training set, which underwent reconstruction.

#### 5.1.1. Extract the Signal Envelope

When bearing failure occurs, it often introduces nonlinear vibration characteristics, such as resonance, impact, or impact failure. Observing the energy distribution in the Teager energy envelope spectrum in Figure 26 reveals variations in peak shape, additional high-frequency components, and nonlinear harmonics.

#### 5.1.2. Envelope Denoising

The Teager energy operator can highlight the nonlinear characteristics and instantaneous energy changes in a signal. Therefore, in CEEMDAN decomposition, the resulting IMF components often better capture the signal’s local features. Additionally, the optimal IMF components may exhibit more pronounced energy concentration and energy leaps. Simultaneously, it can enhance the signal’s time–frequency localization properties. Consequently, the optimal IMF components may be more localized in both the time and frequency domains, aligning more closely with the actual signal’s local characteristics. Figure 27 shows the optimal IMF components of the envelope of an actual bearing signal.

#### 5.1.3. Signal Reconstruction

Figure 28 shows the actual bearing signal after signal reconstruction. It can be seen that the fault impact of the signal is more obvious and that the noise is significantly reduced compared with the original signal.

As can be clearly seen, the fault features of the signal are more prominent after reconstruction. This method highlights the degradation points of the bearing, improving the accuracy of the prediction network in identifying these degradation points. Consequently, it aims to enhance the overall predictive accuracy.

### 5.2. Time–Frequency Optimization

To transform the enhanced time domain signal into the time–frequency domain, a generalized S-transform was applied to this signal. Figure 29 illustrates the time–frequency spectra of the reconstructed signal for different values of *p*, and Table 3 presents the energy values of the time–frequency spectra for various *p* values.

As seen in Figure 29 and Table 3, the optimized time–frequency spectra obtained through the SO algorithm exhibit more pronounced fault features, with higher energy values in the time–frequency spectra.

### 5.3. Bearing Life Prediction

Each sampled data point underwent signal reconstruction and optimization using the SO algorithm to obtain the optimal time–frequency spectrum. All optimal time–frequency spectra were then compiled into an input sample set. These data were fed into the DenseNet-ALSTM model for training.

#### 5.3.1. PHM-2012 Bearing Dataset

During training, bearings 1-7 and 2-5 were selected as the test set, while the remaining bearings were used for training. Bearing 1-7 was sampled 2259 times with a lifespan duration of 22,590 s, and bearing 2-5 was sampled 2311 times with a lifespan duration of 23,110 s. The time–frequency spectrogram images of these two groups of bearings were labeled, assigning a label to each image. The first image was labeled as 1, and so on, with the *i*th image representing the time–frequency spectrogram of the *i*th sampling. The ratio of the duration between the moment corresponding to the *i*th spectrogram and the bearing failure time to the overall duration from the beginning of the bearing to the bearing failure time is referred to as the remaining useful life of the bearing (*y_i_*). The formula for *y_i_* is given by [28,29,30]
(16)$$y_{i}=\frac{n-i}{n-1}$$
where *n* represents the total number of time–frequency spectra in this dataset, and *i* is the label or index of the current moment’s time–frequency spectrum.

The DenseNet-ALSTM network used in this study is primarily composed of three parts: the DenseNet network, attention mechanism, and LSTM. Its input consists of the preprocessed optimal generalized S-transform time–frequency spectrogram images. To validate the effectiveness of this approach, the predicted results obtained through the proposed method were compared with the predictions made by the DenseNet-ALSTM network on the original time domain signals, as well as the predictions from the DRN-BiGRU and CNN-BiGRU networks. Finally, sufficient ablation experiments were conducted to validate the performance of each network module.

Figure 30 shows the prediction results for bearing 1-7, and Figure 31 shows the prediction results for bearing 2-5.

The comparison among Figure 31a–c in the above three figures indicates that the predicted results of the network proposed in this study are more accurate compared to both the DRN-BiGRU and CNN-BiGRU networks, which are two commonly used network architectures. Additionally, when comparing Figure 31a,c,f, it is evident that the proposed optimal time–frequency spectrogram method leads to a significant improvement in prediction accuracy. Furthermore, through comparative analysis in ablation experiments, it was found that the network modules proposed in this paper play a crucial role in enhancing accuracy.

To more intuitively demonstrate the advantages of time–frequency spectrogram prediction, the accuracy of the method is assessed using MAE (Mean Absolute Error), maximum error, and RMSE (Root Mean Square Error) metrics. Table 4 presents the comparative results of the prediction errors.

From Table 4, Compared to other methods, the bearing remaining useful life prediction approach proposed in this paper, which combines the optimal time–frequency spectrum with the DenseNet-ALSTM network, demonstrates higher accuracy and superior performance.

#### 5.3.2. XJTU-SY Rolling Bearing Dataset

To further validate the effectiveness of the proposed method, this study once again utilizes the XJTU-SY bearing full lifecycle dataset to predict the remaining useful life of the bearings. Bearing 1-5 is chosen as the test set, while the remaining bearing data are used for training the network. Similar to the comparison method used in the PHM-2012 dataset, comparisons are made with two commonly used networks, the DRN-BiGRU and CNN-BiGRU networks. Additionally, ablation experiments are conducted to showcase the superiority of the proposed method. As shown in Figure 32, the prediction results for bearing 1-5 are depicted.

The comparison between Figure 32 and Table 5 clearly indicates a significant reduction in errors when using the method proposed in this paper compared to other approaches. The experiments conducted with two different datasets also demonstrate the superiority of the proposed method.

## 6. Conclusions

During the operation of rolling bearings, they are often susceptible to interference, such as background noise, making it challenging to identify and extract fault pulse characteristics. This difficulty in feature recognition can lead to inaccuracies in the life prediction process, resulting in significant errors. Addressing this issue, this paper proposes a bearing life prediction method that combines the optimal time–frequency spectrogram with the DenseNet-ALSTM network. The main conclusions are as follows:(1)This paper proposes methods based on signal reconstruction and SO for the optimal time–frequency spectrogram. By employing signal reconstruction, noise is suppressed without altering the time–frequency characteristics of the signal. Subsequently, the SO algorithm is utilized to obtain the most information-rich optimal time–frequency spectrogram for fault features. This method demonstrates strong anti-interference capabilities.(2)This paper proposes a life prediction method that combines the optimal time–frequency spectrogram with the DenseNet-ALSTM network. This approach effectively eliminates irrelevant features from the data, resulting in more accurate prediction accuracy.

## Figures and Tables

**Figure 1 sensors-24-01497-f001:**
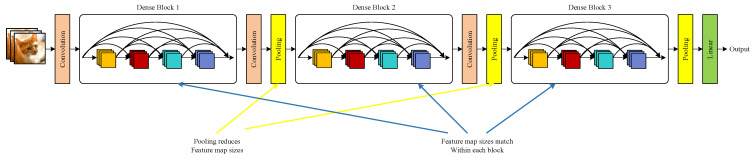
Structure of DenseNet network.

**Figure 2 sensors-24-01497-f002:**
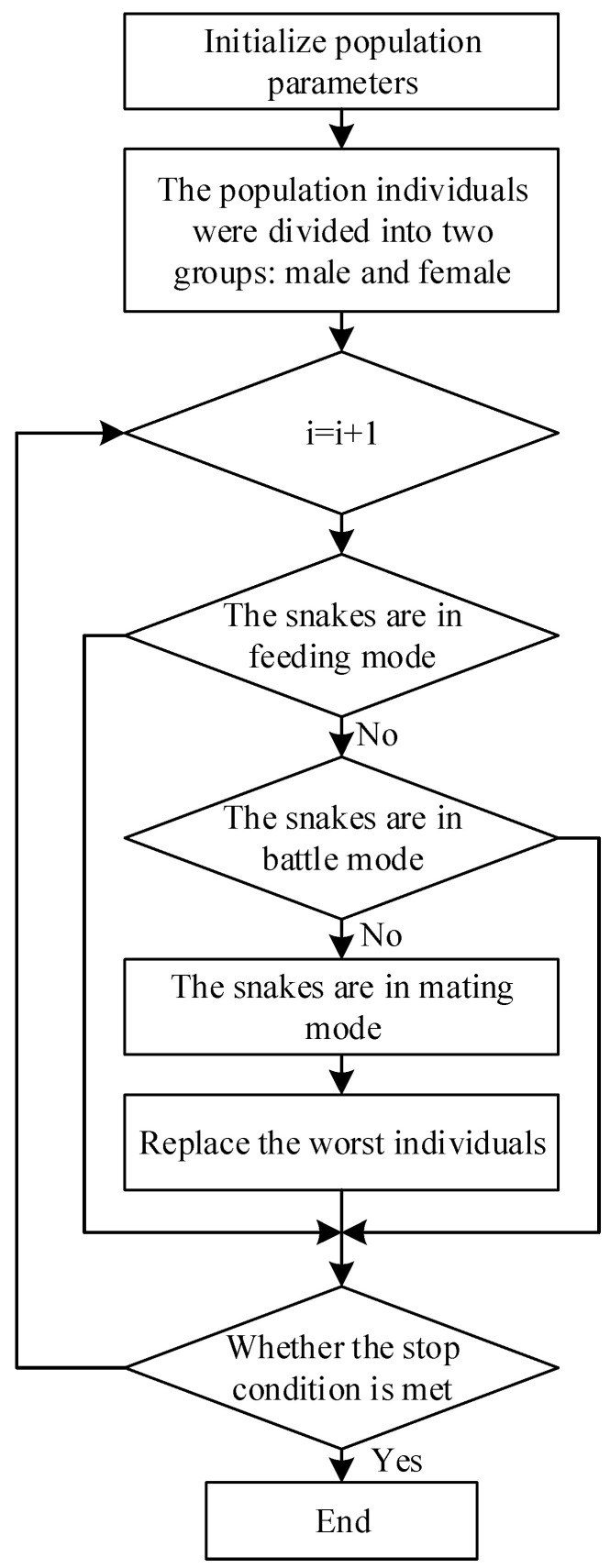
SO algorithm flow chart.

**Figure 3 sensors-24-01497-f003:**
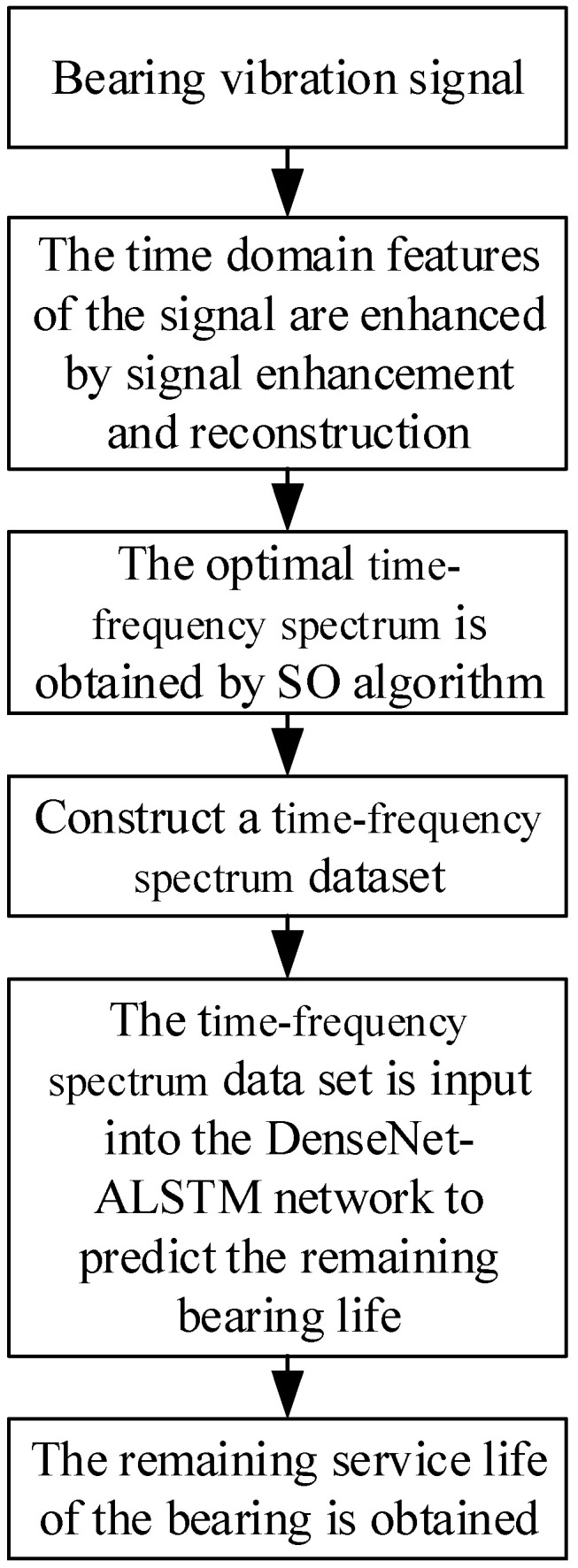
Life prediction of rolling bearing based on optimal time–frequency spectrum and DenseNet-ALSTM flow chart.

**Figure 4 sensors-24-01497-f004:**
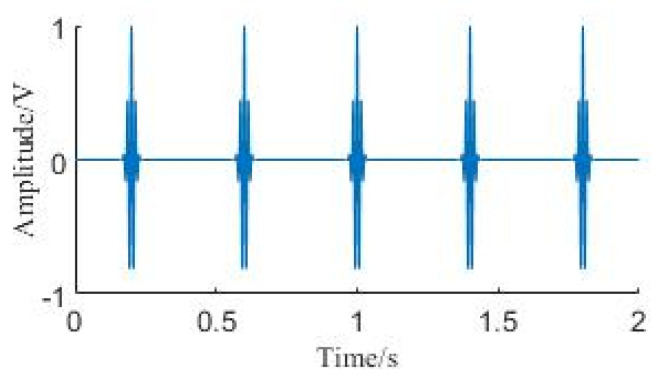
Time domain diagram of the simulated signal.

**Figure 5 sensors-24-01497-f005:**
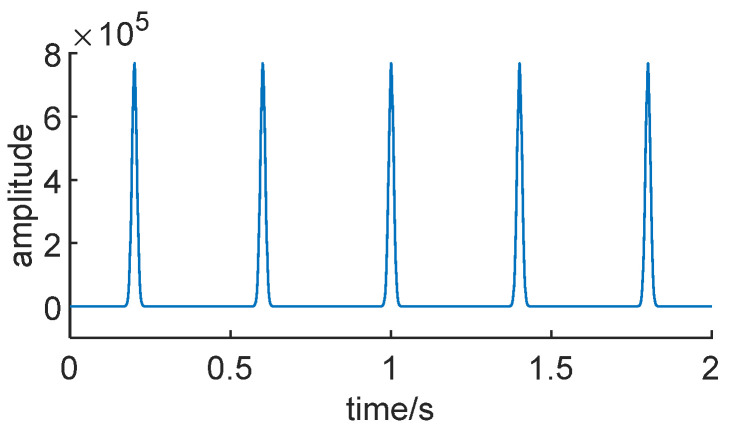
Teager energy operator envelope for signal *x*(*t*).

**Figure 6 sensors-24-01497-f006:**
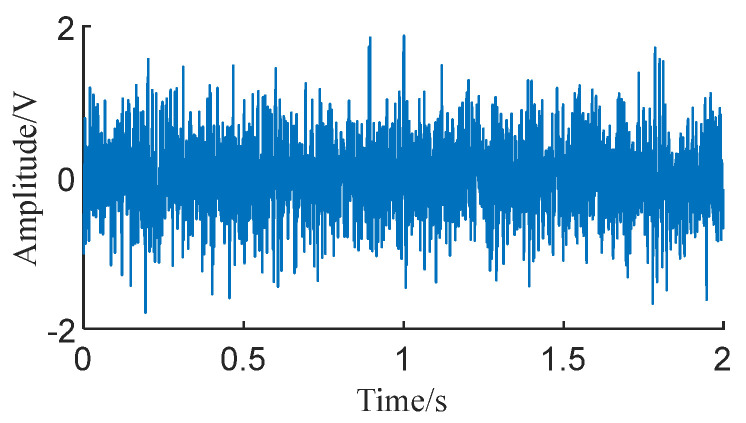
Signal *X*(*t*) time domain diagram.

**Figure 7 sensors-24-01497-f007:**
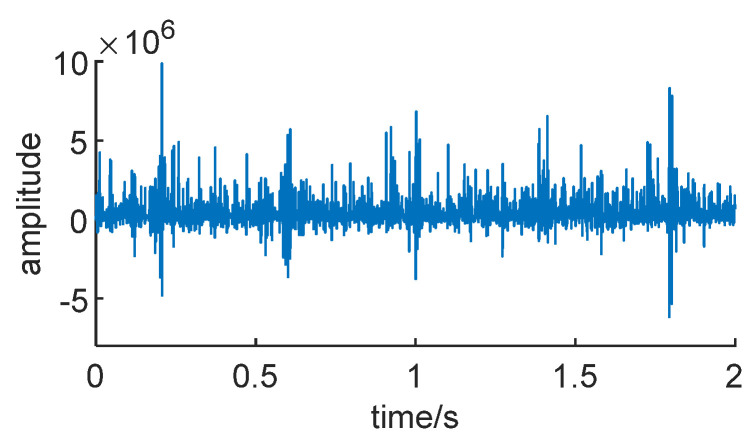
Teager energy operator envelope for signal *X*(*t*).

**Figure 8 sensors-24-01497-f008:**
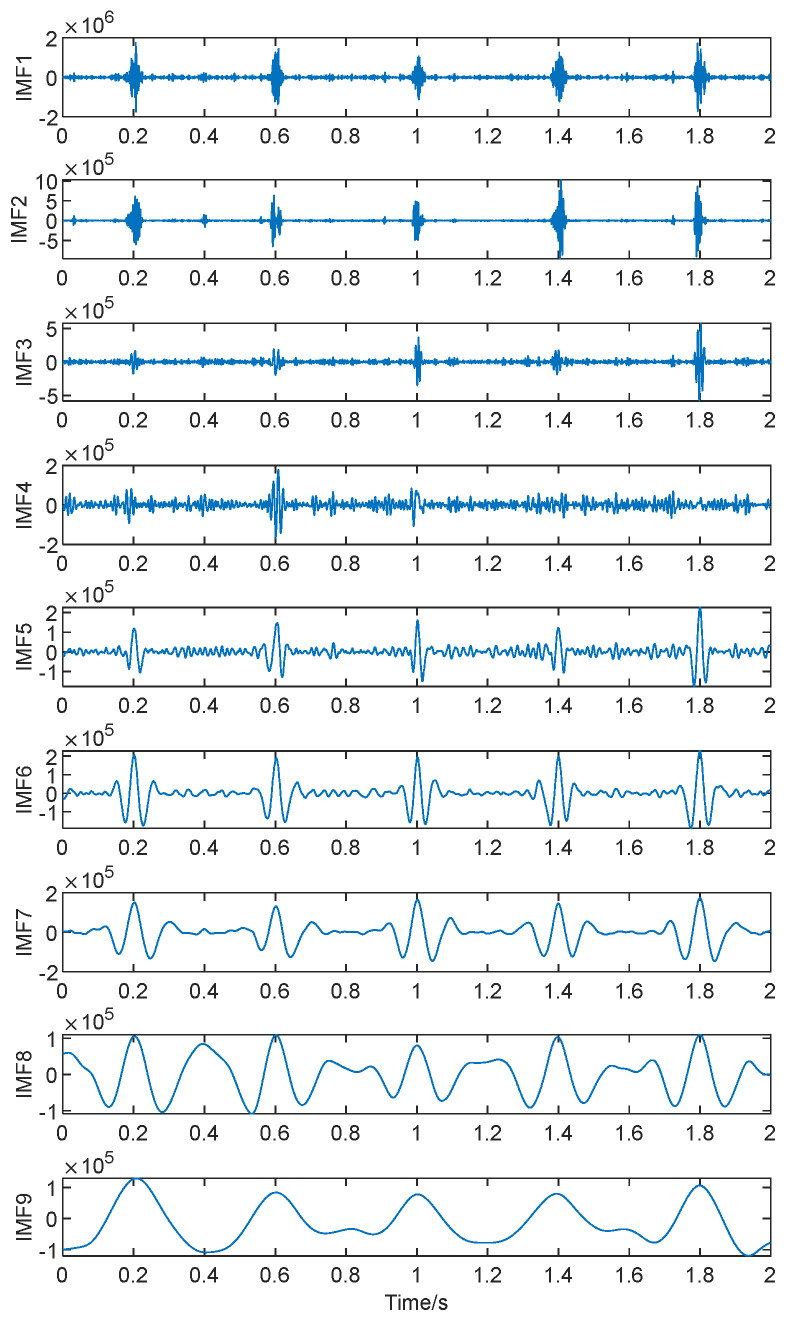
Signal *X*(*t*) envelope CEEMDAN decomposed signal.

**Figure 9 sensors-24-01497-f009:**
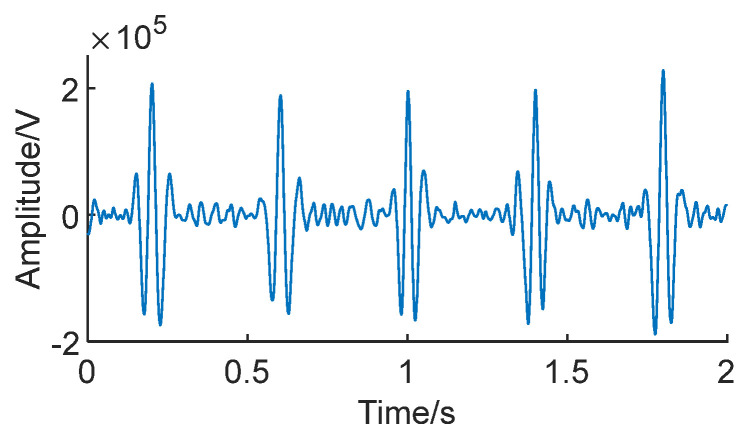
The IMF component with the highest correlation.

**Figure 10 sensors-24-01497-f010:**
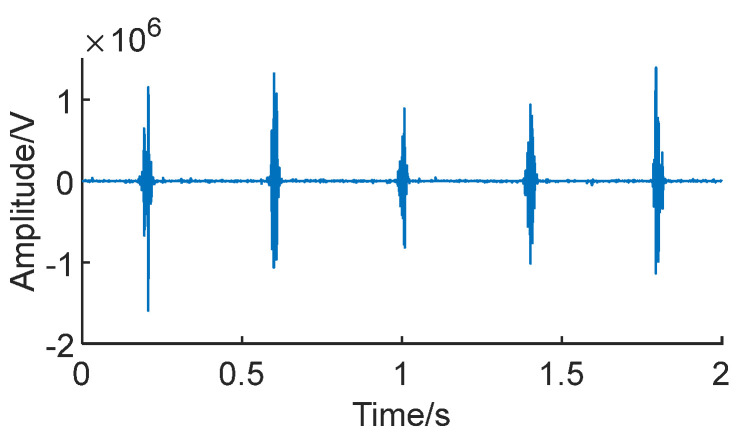
Reconstructed simulation signal.

**Figure 11 sensors-24-01497-f011:**
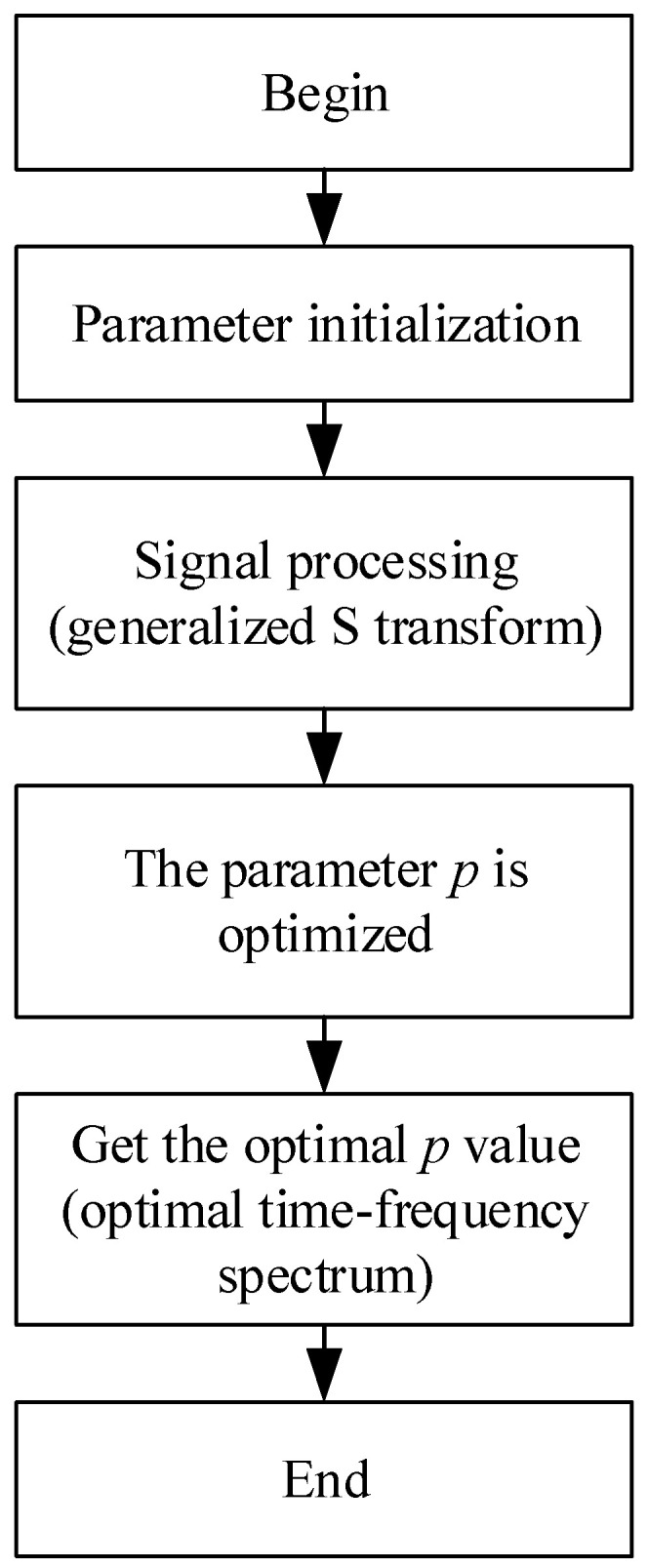
Time–frequency optimization flow chart.

**Figure 12 sensors-24-01497-f012:**
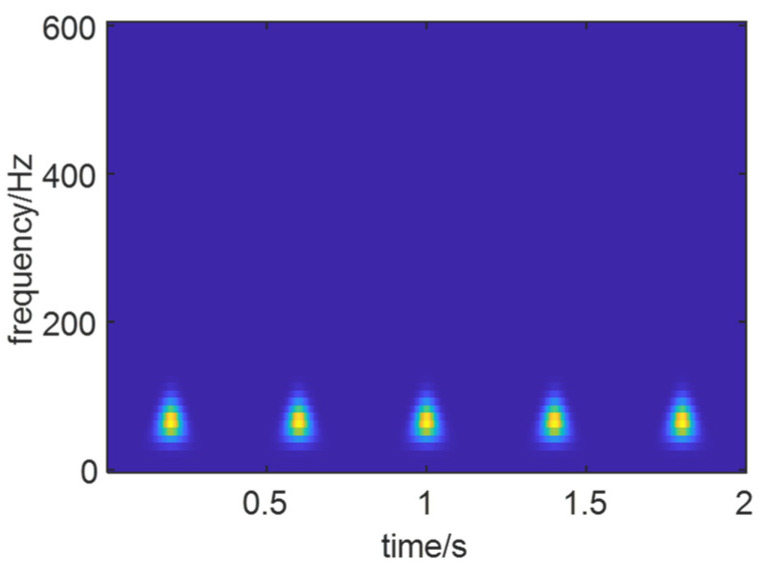
Optimal time–frequency of noiseless simulation signal.

**Figure 13 sensors-24-01497-f013:**
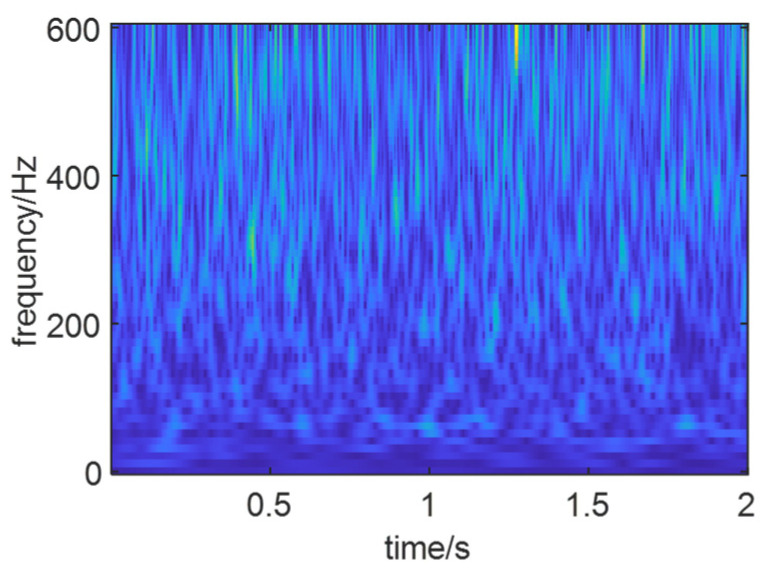
Optimal time–frequency image of signal *X*(*t*).

**Figure 14 sensors-24-01497-f014:**
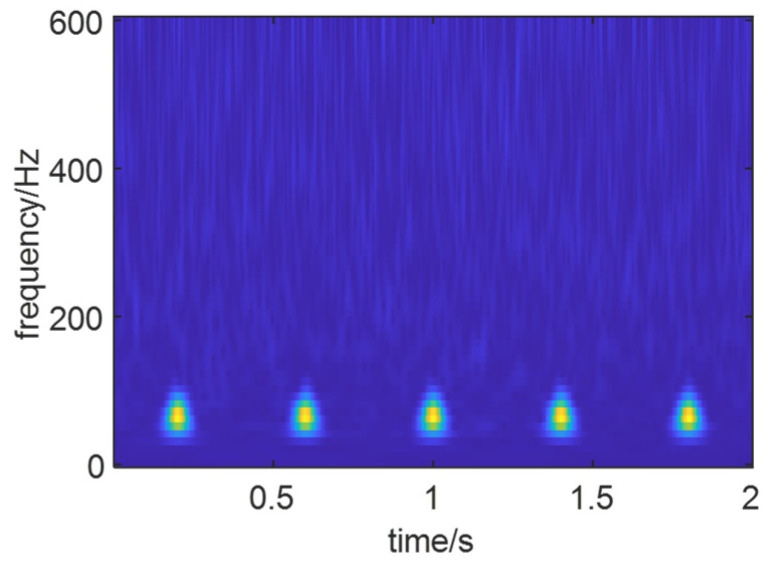
The optimal time–frequency of the reconstructed signal.

**Figure 15 sensors-24-01497-f015:**
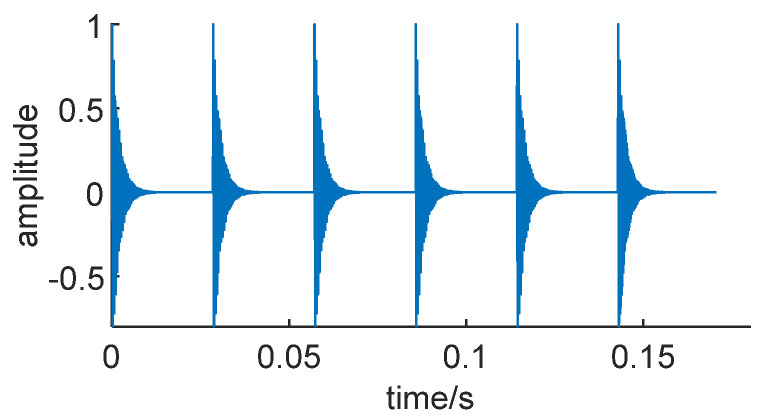
Time domain diagram of bearing fault simulation signal.

**Figure 16 sensors-24-01497-f016:**
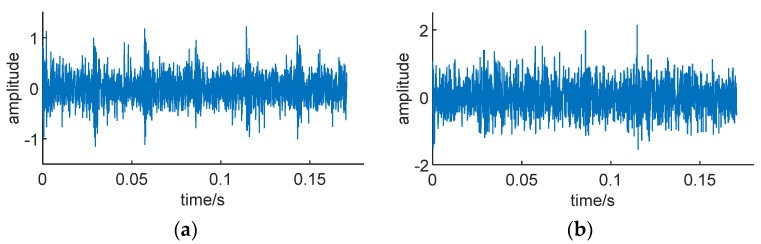
Simulation signal with noise. (**a**) *S*_1_, SNR = −5 db. (**b**) *S*_2_, SNR = −10 db.

**Figure 17 sensors-24-01497-f017:**
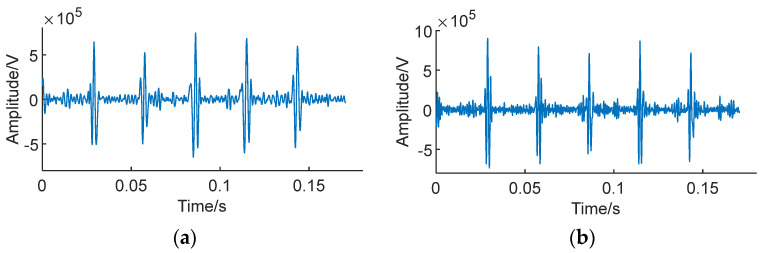
The optimal IMF component after CEEMDAN decomposition. (**a**) *S*_1_, SNR = −5 db. (**b**) *S*_2_, SNR = −10 db.

**Figure 18 sensors-24-01497-f018:**
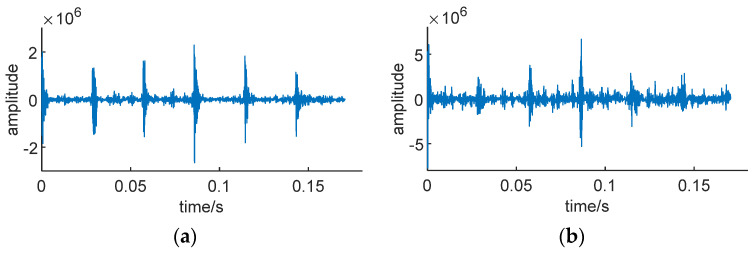
Enhanced simulation signals. (**a**) *S*_1_, SNR = −5 db. (**b**) *S*_2_, SNR = −10 db.

**Figure 19 sensors-24-01497-f019:**
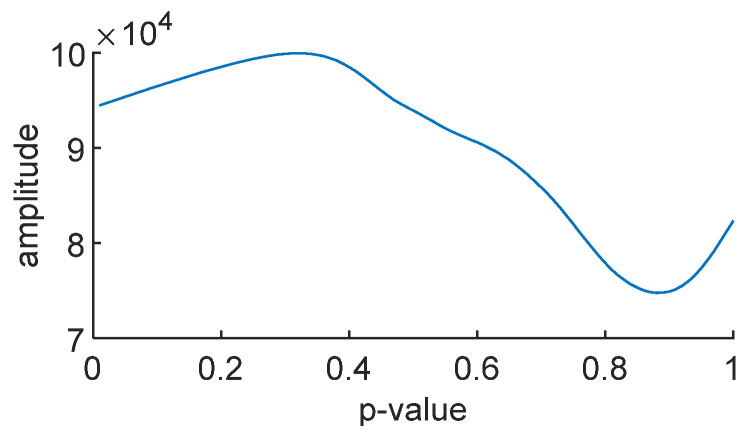
The curve of time–frequency concentration.

**Figure 20 sensors-24-01497-f020:**
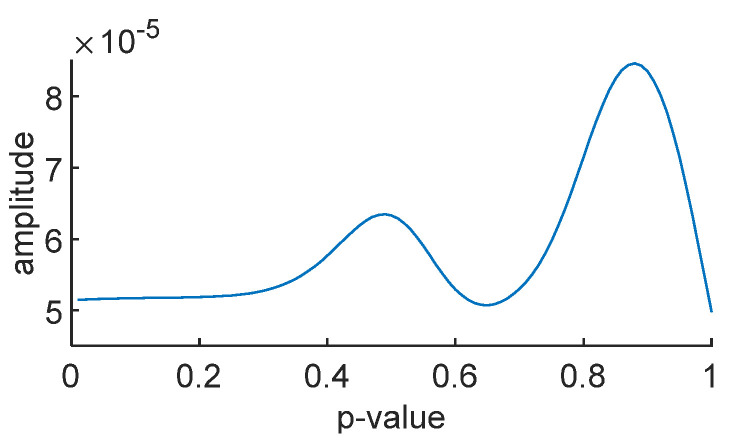
The curve of average energy.

**Figure 21 sensors-24-01497-f021:**
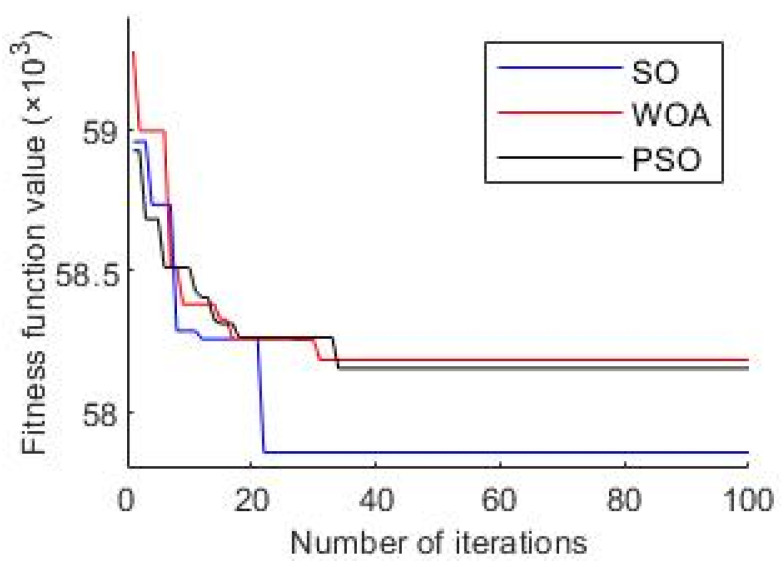
Comparison of optimization algorithms.

**Figure 22 sensors-24-01497-f022:**
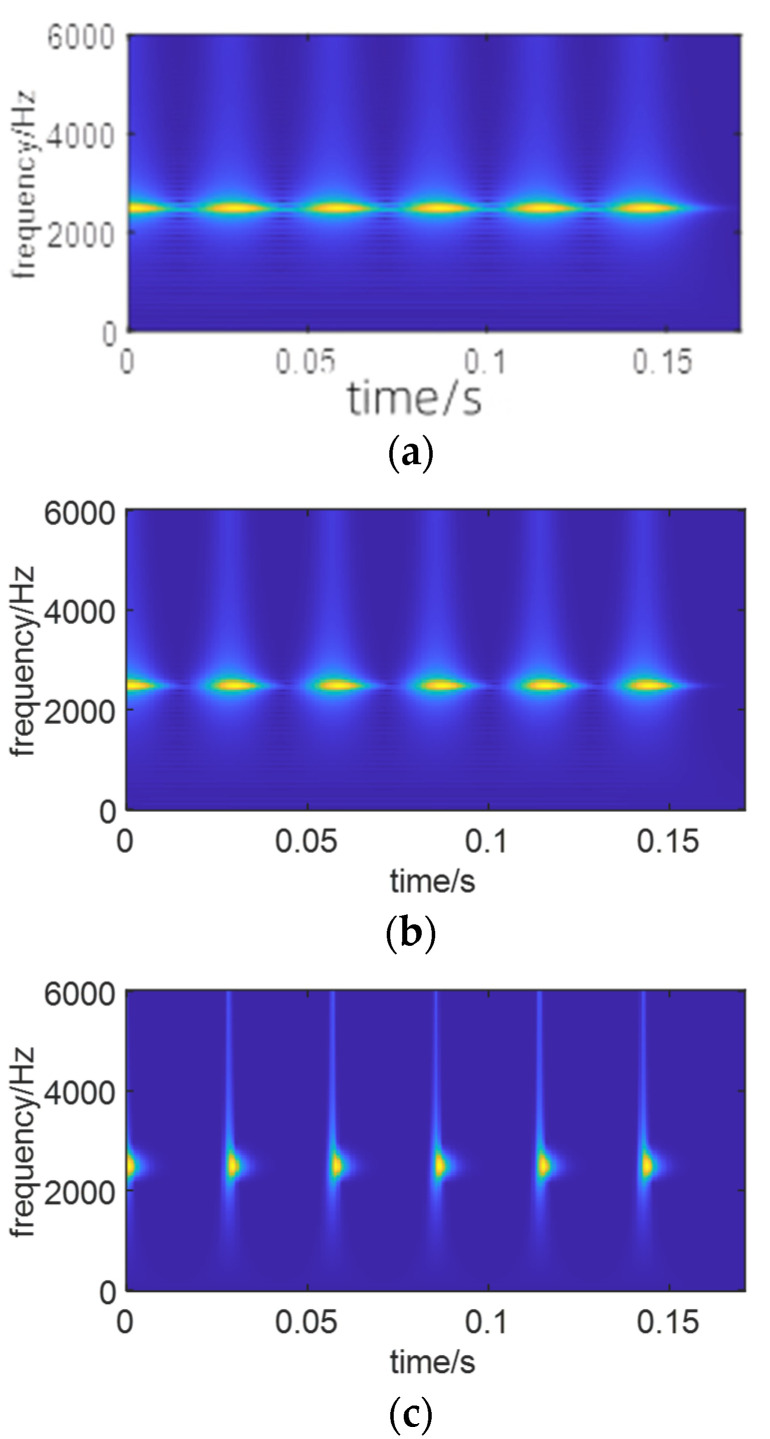
The optimal time–frequency is obtained by three optimization algorithms. (**a**) WOA, *p* = 0.6558. (**b**) PSO, *p* = 0.6667. (**c**) SO, *p* = 0.8865.

**Figure 23 sensors-24-01497-f023:**
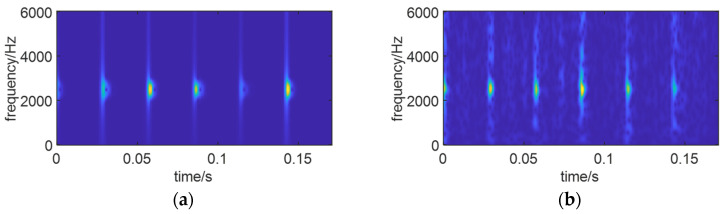
Optimal time–frequency spectra. (**a**) *S*_1_, SNR = −5 db. (**b**) *S*_2_, SNR = −10 db.

**Figure 24 sensors-24-01497-f024:**
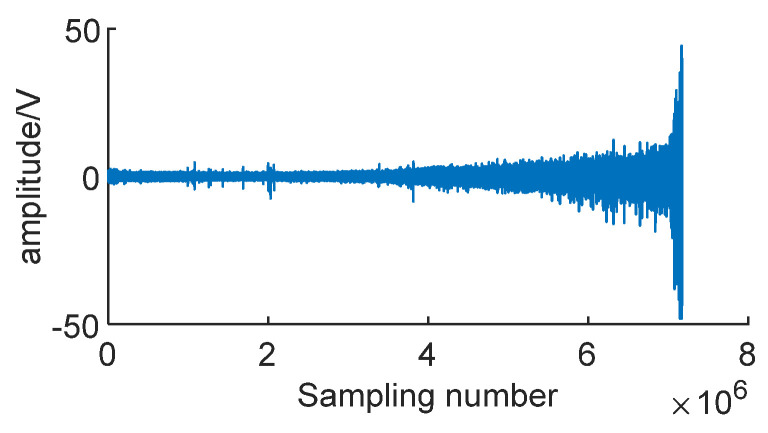
Signal of bearing 1-1.

**Figure 25 sensors-24-01497-f025:**
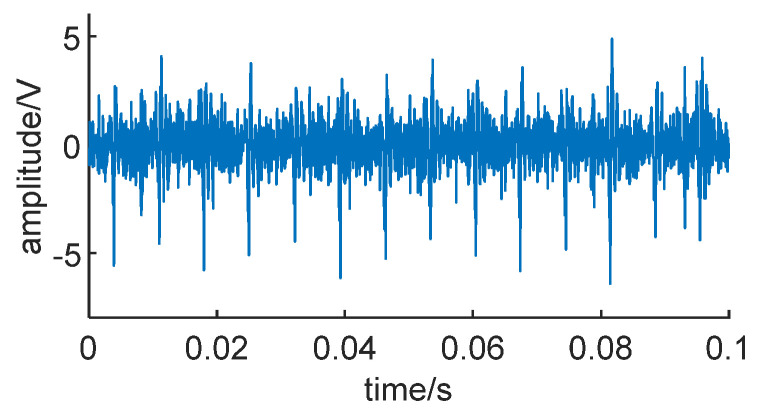
Signal of bearing 3-3.

**Figure 26 sensors-24-01497-f026:**
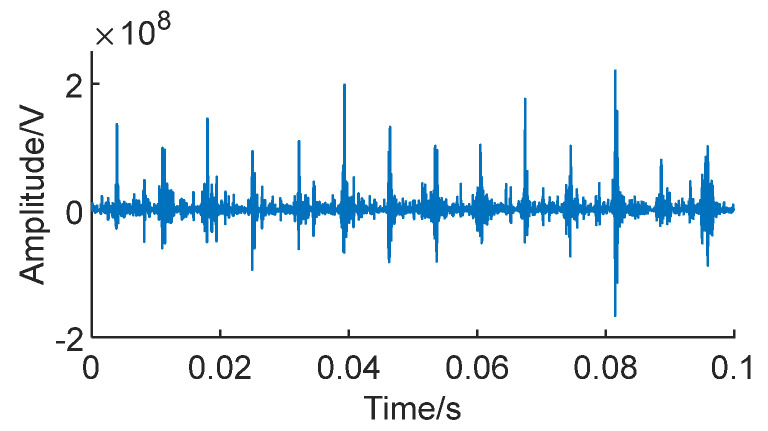
Teager energy operator envelope.

**Figure 27 sensors-24-01497-f027:**
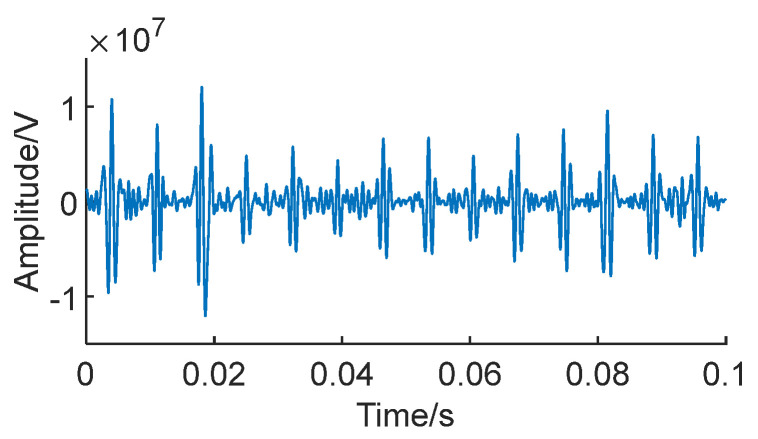
Optimal IMF weight.

**Figure 28 sensors-24-01497-f028:**
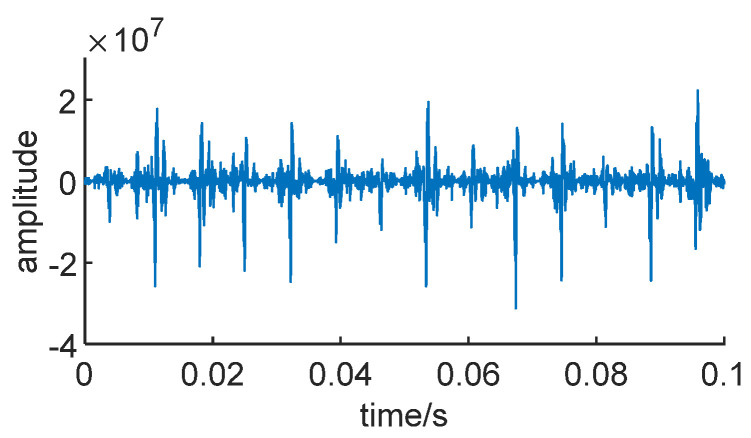
Reconstructed signal.

**Figure 29 sensors-24-01497-f029:**
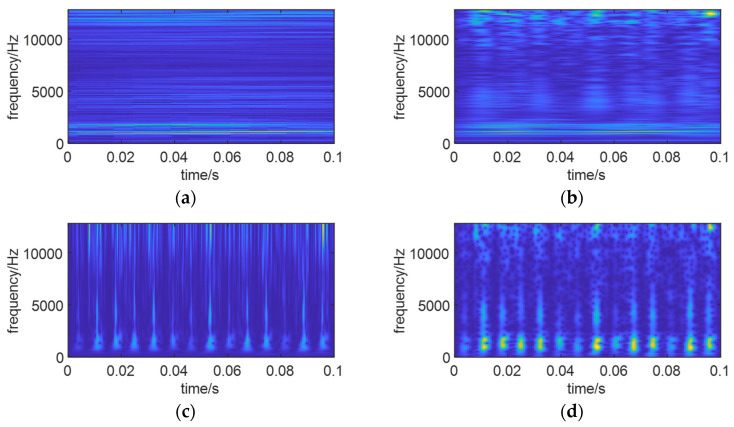
Time–frequency spectra with different values of *p*. (**a**) *p* = 0.3. (**b**) *p* = 0.6. (**c**) *p* = 1. (**d**) Optimizes time–frequency spectrum of SO (*p* = 0.833,9).

**Figure 30 sensors-24-01497-f030:**
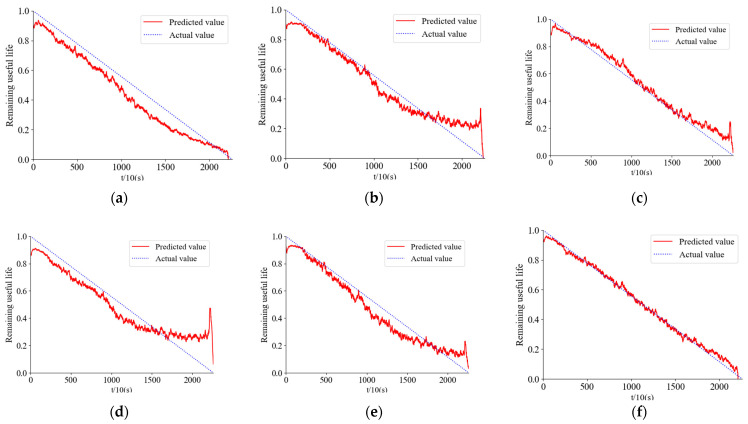
Bearing 1-7 prediction results. (**a**) DenseNet-ALSTM network time domain signal prediction. (**b**) DRN-BiGRU network time domain signal prediction. (**c**) Prediction by CNN-BiGRU. (**d**) Prediction by LSTM. (**e**) Prediction by ALSTM. (**f**) Prediction by this method.

**Figure 31 sensors-24-01497-f031:**
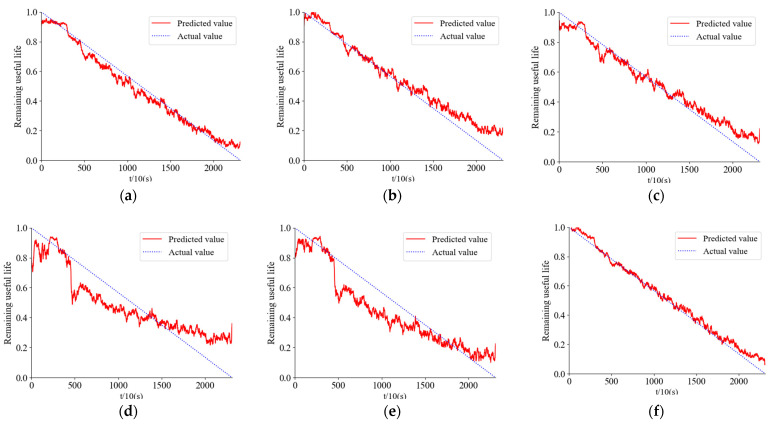
Bearing 2-5 prediction results. (**a**) DenseNet-ALSTM network time domain signal prediction. (**b**) DRN-BiGRU network time domain signal prediction. (**c**) Prediction by CNN-BiGRU. (**d**) Prediction by LSTM. (**e**) Prediction by ALSTM. (**f**) Prediction by this method.

**Figure 32 sensors-24-01497-f032:**
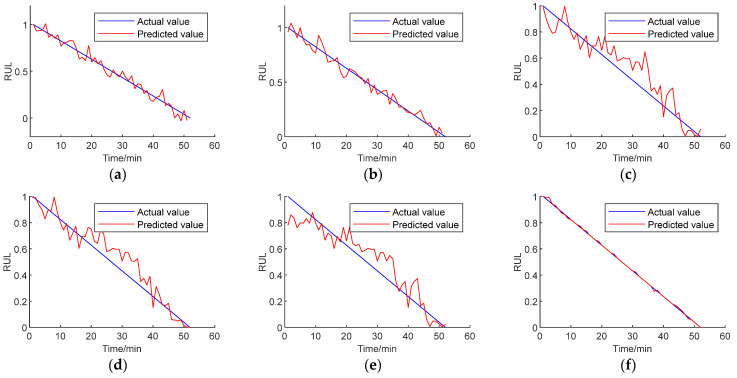
Bearing 1-5 prediction results. (**a**) DenseNet-ALSTM network time domain signal prediction. (**b**) DRN-BiGRU network time domain signal prediction. (**c**) Prediction by CNN-BiGRU. (**d**) Prediction by LSTM. (**e**) Prediction by ALSTM. (**f**) Prediction by this method.

**Table 1 sensors-24-01497-t001:** Comparison of MSE index of three denoising methods.

Author	Before Denoising	After Denoising
Zhao	7.42%	0.90%
Zhuang	6.55%	0.88%
Meng	8.90%	0.75%

**Table 2 sensors-24-01497-t002:** Comparison of SNR before and after signal reconstruction.

	Before Refactoring	After Reconstruction
*S* _1_	−5 db	4.17 db
*S* _2_	−10 db	−0.32 db

**Table 3 sensors-24-01497-t003:** Time–frequency spectrum energy with different values of *p*.

*p*-Value	Energy Value
0.3	1.247, 6
0.6	1.729, 9
1	1.661, 6
0.8339 (optimum)	2.026, 9

**Table 4 sensors-24-01497-t004:** Prediction error comparison.

Bearing Number	Error Type	DenseNet-ALSTM Time Domain Signal	DRN-BiGRU Time Domain Signal	CNN-BiGRU	LSTM	ALSTM	The Method in this Paper
1-7	MAE	0.0566	0.0986	0.0878	0.2356	0.2146	0.0111
	Maximum error	0.0988	0.2288	0.2122	0.4026	0.3365	0.0233
	RMSE	0.0923	0.2132	0.2011	0.3698	0.3025	0.0263
2-5	MAE	0.0860	0.1218	0.1156	0.3569	0.2029	0.0255
	Maximum error	0.1313	0.2320	0.1279	0.3987	0.3296	0.0777
	RMSE	0.1200	0.1756	0.1214	0.3566	0.2989	0.0520

**Table 5 sensors-24-01497-t005:** Prediction error comparison.

Bearing Number	Error Type	DenseNet-ALSTM Time Domain Signal	DRN-BiGRU Time Domain Signal	CNN-BiGRU	LSTM	ALSTM	The Method in this Paper
1-5	MAE	0.1457	0.0963	0.0944	0.2247	0.2368	0.0106
	Maximum error	0.0965	0.1369	0.2008	0.3598	0.3565	0.0133
	RMSE	0.1025	0.1298	0.2149	0.3666	0.3152	0.0128

## Data Availability

The data are available from the corresponding author on reasonable request.

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
