# Peer review of "Life Prediction of Rolling Bearing Based on Optimal Time–Frequency Spectrum and DenseNet-ALSTM"

_sensors, 2024, doi:10.3390/s24051497_

Round 1

Reviewer 1 Report

Comments and Suggestions for Authors

This paper proposed a method based on optimal time-frequency spectra and the DenseNet-ALSTM network to predict the remaining useful life (RUL) of the rolling bearing.

The proposed method used the CEEMDAN deconvolution method to reconstruct the original signal, then the Snake Optimizer was used to obtain the optimal time-frequency spectra. All mentioned signal processing methods were verified in the simulation signal. Finally, all processed data was input into the DenseNet-ALSTM network for life prediction, achieved good results in the PHM-2012 bearing full-life dataset. However, there are some problems below:

1.       Figure 1. is too simple to introduce the structure of the network, there should be more information about the network in the structure figure.

2.       There is detailed introduction of the data pre-processing of the simulation signal. However, the introduction of the data pre-processing of the signal in the dataset is too brief, the optimization process has been completely omitted.

3.       The citation of the dataset in line 540 is incorrect. Please cite the paper mentioned in the IEEE PHM2012-Challenge-Details.

4.       The ablation experiment is inadequate. There is only the comparison between DenseNet-ALSTM network time domain signal prediction and the proposed method, more ablation experiments are needed to confirm the role of each proposed module. Also, it would be better if there were more comparisons between the frequently-used networks and the proposed method. 

5.       There is only one dataset to verify the proposed method, the algorithm needs to be validated on more datasets.

Comments on the Quality of English Language

Language could be improved.

Author Response

Dear Reviewer,

Thank you for your careful review of our paper and for providing valuable feedback. We have now revised the paper according to the suggested modifications.

  1. The authors complete and extend the detailed network structure in Figure 1.
  2. The author supplements the signal preprocessing part of the actual data set, and completes the optimization process of omission.
  3. The author has revised the quoted article.
  4. The authors completed a supplementary experiment and added a comparison experiment of common networks.
  5. The experimental results of XJTU -SJ dataset are added in the experimental part.

Reviewer 2 Report

Comments and Suggestions for Authors

- In the abstract and in the introduction, it should be made clear that the terms used optimization in various contexts are due to the optimization method and state which one. Then it will be clear that the term used has its reference in the research method.

- The authors do not consider the issue of unevenness of the rolling surfaces in the bearing (especially the solid raceway) at all. The magnitudes of the inequalities change with the operating process. They can, for different rotations, induce dynamic couplings with the elements that perform rotational motion (I refer to the Mandelstam criterion).

- The process defined noise does not take into account the possibility of vibrations coming from treadmill irregularities.

- The article lacks publications by authors who have dealt with similar phenomena occurring in rolling bearings, such as scientists from Europe.

Author Response

Dear Reviewer,

Thank you for your careful review of our paper and for providing valuable feedback. We have now revised the paper according to the suggested modifications.

  1. Thank you for pointing this out. In the abstract and introduction sections, it will be explicitly stated that the optimization terminology used in different contexts is due to the adoption of the CEEMDAN method. This will clearly indicate the reference nature of the terms used in the research methodology.
  2. In the Mandelstam criterion, when sound waves propagate through it, small defects (such as cracks or voids) will cause scattering and attenuation of the sound waves. According to the Mandelstam criterion, this sound wave attenuation is related to the size of the defect and the properties of the material. Typically, larger defects and softer materials lead to greater attenuation effects. However, the rotational indicators used in the bearing dataset in this paper, such as speed, are the same. There is no dynamic coupling problem between the rotational indicators of the training set and the test set. Furthermore, the bearing models are identical, and the impact of the unevenness in experimental results can be negligible.
  3. The noise defined in the simulated signals in this paper is Gaussian white noise, and the experimental results are good. However, in the bearing life prediction experiments, the background noise in the dataset includes irregular vibrations similar to those from a treadmill, which has been taken into account and yields good results.
  4. Relevant research papers on bearing life prediction by European Norwegian scientists have been added.

Reviewer 3 Report

Comments and Suggestions for Authors

The paper deals with a DenseNet-ALSTM network for life prediction of identifying a bearing failure. The authors used the data to verify their theory. And the scope of the data group can be discussed, but I consider it sufficient for the initial verification of the presented method. The results of the paper are defined and aplly. The form of the paper is understandable. I accept the paper in the presented form.

Author Response

Dear Reviewer,

Thank you for reviewing our paper and agreeing to accept it. We greatly appreciate the valuable feedback you have provided, which has played a crucial role in improving the quality of the paper.

Round 2

Reviewer 1 Report

Comments and Suggestions for Authors

No further question.

Author Response

Dear Reviewer,

Thank you very much for reviewing our paper and for agreeing to accept it for publication. Your approval means a lot to us. We will carefully consider the suggestions you provided and make appropriate modifications and additions in the final version to ensure that our research reaches the highest standards.

Once again, thank you for your support and valuable feedback. We look forward to continuing our collaboration in future exchanges.

Reviewer 2 Report

Comments and Suggestions for Authors

The authors' answer regarding the Mandelstam criterion is questionable, to say the least. Because this criterion deals with dynamic coupling between two systems and is related to the frequencies involved. If we label one system as 1 and the other as 2 then if W1 to W2 is much less than unity then there is no dynamic coupling between the systems. On the other hand, if W1 to W2 is near unity then the two systems can generate dynamic coupling. With regard to bearings, this principle can apply to two systems: a system of bearing inequalities generating frequencies W1 and a system of flexural vibrations of rotating elements or vibrations caused by systems connected to rotating elements, which is connected to a bearing having an eigenfrequency W2 If this ratio is much greater than unity, and this is most often the case in a system of rolling bearing and element connected to a bearing, then there will be no dynamic coupling. The authors may or may not include this issue in the article.

Of course, the issue of bearing reliability is also being addressed by researchers in Germany, Sweden and Poland. To my knowledge, these works are not compatible with the article.

Author Response

Dear Reviewer,

Thank you for your review of our paper and for providing valuable insights into the Mandelstam criterion. Regarding your point about the dynamic coupling between W1 and W2, we understand that this involves the frequency relationship between two systems, particularly within the context of bearing systems where one system generates frequencies labeled as W1 and the other system produces inherent frequencies labeled as W2. If W1 to W2 is significantly less than unity, there is no dynamic coupling between the two systems. Conversely, if W1 to W2 approaches unity, the two systems may exhibit dynamic coupling.

We would like to emphasize that our research primarily focuses on bearing data and simulation experiments, without considering the effects of resonance phenomena at the mechanical level. Within the scope of our study, we did not take into account the relationship between W1 and W2 and unity, as our assumptions were based on not considering this frequency dynamic coupling. We will supplement and emphasize this limitation in the paper.

Regarding studies from Germany, Sweden, Poland, and other countries, we are very interested and will carefully examine these references to ensure that our work has sufficient comparison and reference in relevant areas. We will incorporate appropriate citations and comparisons in the final paper to ensure that our research presents a comprehensive and accurate contribution to bearing reliability.

Thank you once again for your review. We will carefully consider your suggestions and make appropriate revisions in the final version of the paper.